# Fast identification of the charging pile plug materials using laser-induced breakdown spectroscopy

Lidan Chen[1], Tiantian Pan [2]*, Liuye Cao[2], Fei Liu[2]

**1** College of Automotive Technology, Zhejiang Technical Institute of Economics, Hangzhou, China,
**2** Zhejiang Key Laboratory of Agricultural Remote Sensing and Information Technology, College of Biosystems Engineering and Food Science, Zhejiang University, Hangzhou, China

* ttpan@zju.edu.cn

## Abstract

The electric vehicles (EVs) is showing rapid growing, with charging piles playing a critical role as essential infrastructure. The performance and reliability of charging plugs directly influence grid efficiency, while conventional copper-based materials present several limitations. Tellurium copper is an alloy well-suited for charging plugs. Identification of the materials is crucial for ensuring the electrical performance and safety, and recycling value. In this study, laser-induced breakdown spectroscopy (LIBS) was utilized for rapid identification of tellurium copper, red copper and brass. The tellurium in the alloy was identified and the LIBS parameters were optimized. K-nearest neighbor (KNN), random forest (RF), and convolutional neural networks (CNN) models were built for discrimination of three kinds of materials. Knowledge-driven feature extraction based on database and two data-driven feature extraction methods, successive projections algorithm (SPA) and competitive adaptive reweighted sampling (CARS), were used to select feature bands. The optimal models achieved accuracy of 100% both for training set and testing set, indicating that the LIBS could realizing the rapid identification of charging plug materials. The proposed LIBS-based identification method helps ensure the safety and reliability of charging stations, support the healthy development of the EV industry.

## Introduction

The global pursuit of carbon neutrality has significantly accelerated the adoption of electric vehicles (EVs), with charging piles playing a critical role as essential infrastructure [1]. The performance and reliability of charging pile plugs directly influence user experience, grid efficiency and charging safety, as these plugs must meet stringent requirements such as high-frequency plugging and unplugging, high-current transmission, and resistance to high temperatures [2]. Conventional copper-based materials, including brass and red copper, are commonly used due to their excellent electrical conductivity; however, they present several limitations and potential

**Data availability statement:** The dataset and code are available from the GitHub repository (https://github.com/ptthoshi-web/LIBS-Charging-pile-plug).

**Funding:** The author(s) received no specific funding for this work.

**Competing interests:** The authors have declared that no competing interests exist.

risks. These materials often lack sufficient mechanical strength, leading to wear and deformation from repeated use [3]. Additionally, during fast charging, the high current can cause significant temperature rise, which may soften the copper and compromise contact stability [4]. Furthermore, red copper's high ductility makes it difficult to process complex structures efficiently, resulting in increased costs and lower manufacturing efficiency [5]. As charging pile plugs require a balance of high conductivity, mechanical strength, corrosion resistance, and ease of processing [6], general materials like brass, beryllium copper, and red copper, while partially meeting these needs, are often associated with issues such as high costs and environmental concerns. This underscores the growing demand for more advanced material solutions.

Tellurium copper (0.4%-0.7% tellurium in copper) is an alloy well suited for charging pile plugs due to its excellent electrical conductivity (≥90% IACS), high mechanical strength, wear resistance and thermal stability [7,8]. These properties help ensure efficient energy transmission, reduce heat generation [9], and extend the service life of the charging plugs, making it ideal for high-current, fast charging applications. Additionally, its enhanced machinability lowers manufacturing costs, while its non-toxic and recyclable nature aligns with sustainability goals [10]. Tellurium copper improves safety by reducing conformity resistance and preventing overheating, ensuring stable performance, and reducing maintenance costs [11]. Widely recognized in industry standards, it is already used by companies like Tesla and is set to play a key role in the future of electric vehicle infrastructure as fast-charging technologies evolve. Identification of the materials is crucial for ensuring the electrical performance and safety of the charging pile plugs. If brass or red copper is mistakenly used in place of tellurium copper, insufficient conductivity or thermal softening may lead to increased contact resistance, overheating of the plugs, and even melting, which could result in fire hazards [12]. Furthermore, red copper has low mechanical strength, making it prone to deformation from frequent plugging and unplugging, leading to poor contact [5]. Brass, with its low wear resistance, may generate metal debris after prolonged use, increasing the risk of short circuits [13]. The superior conductivity and high temperature resistance of tellurium copper [14] make it essential for high-power fast charging applications, which other materials cannot effectively support. Furthermore, tellurium copper plugs are designed to withstand tens of thousands of insertions, whereas brass plugs may fail after only a few thousand cycles. Tellurium copper is more expensive than ordinary brass, and the purity of red copper directly impacts its cost. The charging plugs can deteriorate over time due to factors like electrical overload, environmental changes, and wear-and-tear, which will pose safety risks. The components of the discarded charging plugs should be determined and properly recycled and recovered, which can help reduce resource waste and minimize e-waste pollution. Therefore, identification of the charging plug materials was significant for ensuring charging efficiency, safety, and protection of the environment.

The traditional chemical detection methods require sample pretreatment and need to damage the plugs, making it time-consuming and limiting the scope of screening [15]. Laser-induced breakdown spectroscopy (LIBS) is a micro-damaging atomic

spectroscopy technique, which analyzes the elemental information of samples by collecting the plasma transition spectral lines generated by laser pulse ablation [16]. LIBS detection does not require complex preprocessing and could analyze the multiple elements simultaneously, therefore being widely used in industry [17], agriculture [18], medicine [19] and other fields recently. LIBS has also been applied in some manufacturing inspections related to EV industry. Smyrek et al. [20] utilized LIBS for post-mortem analysis of lithium concentration in electrochemically cycled Li(NiMnCo)O2 cathodes based on calibration studies with electrodes at different State-of-Charges. Kang et al. [21] used optical emission spectroscopy for measurement of laser-induced plasma of aluminum and copper dissimilar laser welding, which is an in-demand process in the manufacture of secondary battery systems for electric vehicles. LIBS has enabled on-site, rapid verification of raw material authenticity, preventing suppliers from substituting inferior materials in EV industry. LIBS performs micro-area ablation and provides in-situ, rapid detection within seconds, making it ideal for real-time monitoring on charging plug production lines. The use of LIBS for rapid detection ensures that incorrect material usage does not lead to premature aging or failure of the charging plugs. By monitoring material consistency during production, LIBS could help reduce batch rework or scrap caused by material errors, thereby lowering overall production costs and ensuring the safety of the plugs during charging. LIBS could also detect the components of the discarded charging plugs, which help recycle metals and promote the sustainable development of the EV industry.

In this study, the LIBS system was utilized for discrimination of three kinds of charging pile plug materials, including tellurium copper, red copper and brass. The aims were: (1) Build machine learning models based on LIBS spectra for classification of three kinds of materials; (2) Identify the spectral lines related to metals that play a key role in classification (3) Enable rapid detection of charging plug material, ensuring optimal performance, preventing safety risks, and supporting the identification and recycling of valuable metals from discarded plugs. The proposed method could ensure the quality of electric vehicle charging infrastructure, contributing to environmental protection and the sustainable growth of the EV industry.

## Materials and methods

### Experimental materials and pretreatment

As the tellurium copper is the materials that mainly investigated in this study, 40 tellurium copper pellets were firstly used as samples for LIBS parameter optimization. Then 60 gold-plated brass pins (H62), 20 gold-plated red copper pins (T2) and 40 tellurium copper pellets (C14500) were samples used for LIBS discrimination model training and evaluation (Fig 1a). The major element composition of three kinds of materials, including copper (Cu), tellurium (Te), zinc (Zn) and other elements, were listed in Table 1. There were gold plating layer on the surface of the brass pins and copper pins, which could possibly affect the LIBS measurement. To explore the strategy to eliminate such influence, the gold plating layer on the surface of 40 brass pin samples were removed by sandpaper. 20 brass pins and 20 red copper pins retained the gold plating layer.

### LIBS setup

A self-assembled LIBS system was used in this study for data acquisition (Fig 1b). The 532 nm pulsed laser was generated by Q-switched Nd: YAG pulsed laser (Vlite-200, Beamtech Optronics, Beijing, China) with a pulse duration of 8 ns and pulse frequency of 1 Hz. The pulsed laser was transmitted through the optical path system and focused by the lens (f = 100 mm) above the sample. The focused pulse laser ablated sample and generated plasma. The distance between lens and sample was optimized to 98 mm. Plasma spectra was collected by two combinations of spectrometers and detectors. An echelle grating spectrometer (ME5000, Andor, Belfast, UK) with the wavelength range of 199-1031 nm and the resolution of 0.03 nm and a monochromator (SR500i, Andor, Belfast, UK) with the wavelength range set to 189-210 nm, and the resolution of 0.02 nm, respectively, were utilized. The signals of two spectrometer were collected by two ICCD detectors (iStar DH334T-18F-03, Andor, Belfast, UK), respectively. A digital delay generator (DG645, Stanford Research

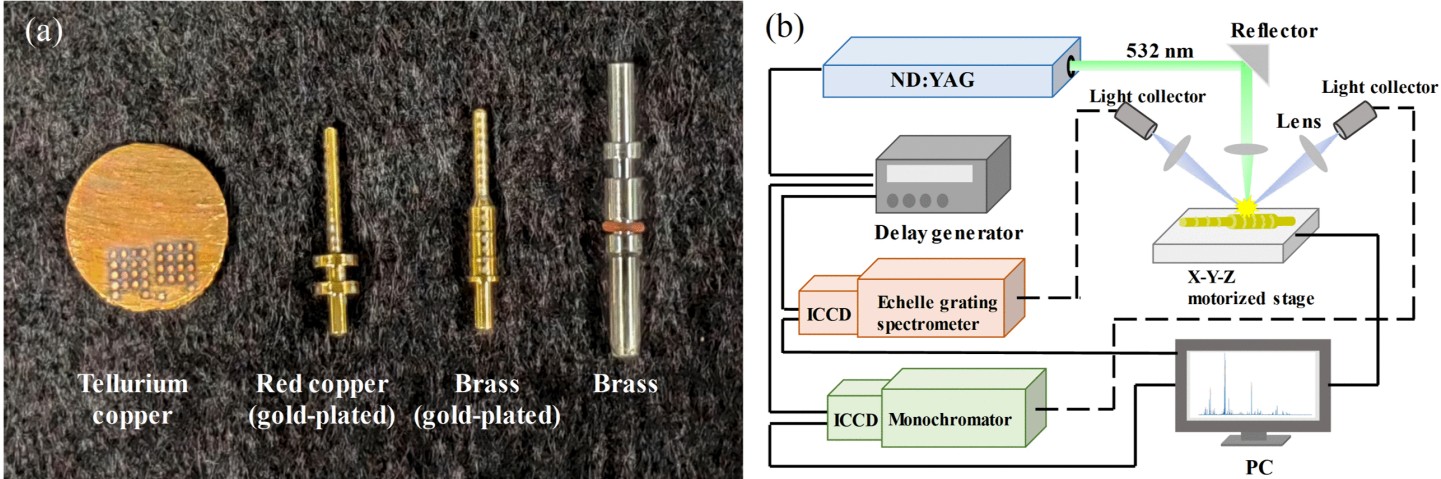

**Fig 1. The experimental materials and instruments.** (a) The photo of tellurium copper pellet, gold-plated red copper pin, gold-plated brass pin, and brass pin with gold-plating layer removed. The samples had been ablated by LIBS. (b) The schematic diagram of LIBS system.

**Table 1**. **The mass fraction of elements in three kinds of materials.**

| Materials | Cu | Zn | Te | Other elements * |
|---|---|---|---|---|
| Tellurium copper | ≥ 99.3% | / | 0.625% | ≤ 0.02% |
| Brass | 60.5% - 63.5% | 36.5% - 39.5% | / | ≤ 1.0% |
| Red copper | ≥ 99.9% | / | / | ≤ 0.1% |

* Other elements including lead (Pb), iron (Fe), nickel (Ni), phosphorus (P), et al.

Systems, California, USA) was used for the sequential control of the laser pulse and the ICCD detectors. An X-Y-Z motorized stage was used to move the sample. For the tellurium copper pellet, a 4×4 array with a spacing of 1 mm was set as the ablative path, and for the charging pile pins, a 1×16 path with a spacing of 1 mm was set. For tellurium copper pellet and brass pins with gold plating layer removed, each location was ablated three times and the signals were accumulated. And for gold-plated brass pins and red copper pins, each location was ablated five times and the signals of the last three times were accumulated, to eliminate the influence of the gold-plating layer. The average spectra of 16 locations were calculated as the spectra of sample to reduce the interference of point-to-point fluctuation and a total of 120 spectra were obtained. The parameters for LIBS spectra acquisition were optimized, and the laser energy was selected from 50, 60, 70, 80 and 90 mJ, the delay time was selected from 1, 1.5, 2, 2.5, 3 μs, and the gate width was set accordingly as 19, 18.5, 18, 17.5 and 17 μs.

### Feature extraction methods

**Knowledge-driven method.** In accordance with the formation mechanism of LIBS spectrum, the LIBS spectral peaks are often fingerprint peaks that specific to particular elements. The elemental information of LIBS can be cross-referenced with databases such as the National Institute of Standards and Technology (NIST) database. Cu is the main element in the three kinds of materials, while Te and Zn were of certain content differences in them (Table 1). Therefore, the spectral lines corresponding to Te and Zn were extracted as feature bands according to the NIST database (Table 2).

**Data-driven methods.** Two data-driven feature extraction methods, successive projections algorithm (SPA) and competitive adaptive reweighted sampling (CARS), were utilized to extracted feature bands for classification of charging plug materials. SPA is a forward iterative search method that begins with a single variable and progressively adds one new

**Table 2**. The spectral lines of Te and Zn extracted from the LIBS spectra acquired by monochromator and echelle grating spectrometer.

| Element | Spectral lines | |
|---|---|---|
| | Echelle grating spectrometer | Monochromator |
| Te | 225.552 nm, 225.903 nm, 226.553 nm, 238.328 nm, 238.579 nm, 253.074 nm | 214.281 nm, 214.725 nm, 215.986 nm, 220.883 nm, 225.552 nm, 225.903 nm, 226.553 nm |
| Zn | 330.258 nm, 334.501 nm, 472.216 nm, 481.053 nm | 213.857 nm |

variable with the maximum projection in each iteration, until the desired number of selected variables is reached [22]. In this study, the number of selected variables was explored from 5 to 50. CARS is a feature extraction method based on partial least squares (PLS) model weighted regression coefficients and features with higher absolute values are selected [23]. The number of CARS features was also explored from 5 to 50.

## Discrimination methods

Principal component analysis (PCA) is an unsupervised classification method commonly used for qualitative analysis. PCA linearly transformed the original data into new orthogonal variables PCs [24]. By plotting the samples with a few PCs, the similarities and differences of them could be represented [25]. In this study, PCA was used to visually demonstrate whether the samples could be grouped and explore the spectral lines and their corresponding elements with high loadings.

Three supervised classification methods, including K-nearest neighbor (KNN), random forest (RF), and convolutional neural networks (CNN), were used in this study for discrimination of the three kinds of samples. To build and estimate the discrimination models, the samples were randomly split to training set and testing set at a ratio of 7:3. The models were trained using the training set, and the classification performance was evaluated using the testing set. KNN is a type of instance-determined method, and the sample class is determined by a majority vote by the classes of the k-nearest neighbor [26]. The number of the nearest neighbors (k) is the important factor influencing the classification accuracy. It was explored from 3 to 20 in this study. RF is an advanced integrated machine learning technique based on decision tree. RF uses bootstrap resampling method to continuously generate training and testing sets, and trains multiple classification trees [27]. The final predictive results are obtained by a majority vote of the predictions of all individual trees. In this study, the number of trees was explored from 1 to 50. The Out-Of-Bag (OOB) error was used as an internal validation measure to estimate model performance. CNN is developed from feed-forward artificial neural network by adding convolution layers. The CNN model used in this study consisted of two convolution layers, and one fully connected layer. The first convolution layer consisted of 16 filters with the size of 2×1. The first convolution layer consisted of 32 filters with the size of 2×1. Both two layers were followed by batch normalization and a ReLU activation function. The Adam optimizer was utilized for training. The training is configured to run for a maximum of 50 epochs. The initial learning rate was set to 0.001, and the learning rate drop factor was set to 0.1. Classification models were built based on feature bands and full spectra.

## Software and model evaluation

The feature extraction and model establishment were implemented on Matlab R2022a (The Math Works, Natick, MA, USA). The performances of the discrimination models were evaluated by the discriminant accuracy of training and testing set. The higher accuracy indicates that the model is highly precise.

## Results and discussion

### Tellurium identification and parameter optimization

The pulse laser ablates the sample surface and induce the plasma. As plasma cools, the kinetic energy of free electrons decreases through collisions with ions, emitting photons that create bremsstrahlung radiation. Compound radiation occurs

when electrons become trapped by ions, forming neutral particles and emitting photons. Both types of radiation produce continuous spectra, contributing to a background continuum. During excitation, electrons transition from high to low energy levels, emitting spectra characteristic of different elements. Initially, the continuum background spectrum is dominant, but as the plasma cools, characteristic radiation becomes more prominent. The laser pulse energy and delay time are important parameters that influence the signal quality of LIBS measurement. The delay time is the period between plasma generation and signal detection, while the integration time is the ICCD gate width. The total of delay and integration times represents the plasma lifetime. The signal quality depends on laser energy as well. If the laser pulse energy is too low, plasma instability can occur due to the sample's ablation threshold [28].

40 copper tellurium pellet samples were used for LIBS parameter optimization. The LIBS spectra of the copper tellurium pellet samples were acquired with laser energy ranging from 50 mJ to 90 mJ and delay time ranging from 1 to 3 μs. For each parameter setting, there were four parallel samples to estimate the stability of LIBS signal. Four Te lines were identified according to the NIST database (Fig 2).

The signal-to-background ratio (SBR) of four Te lines, and the relative standard deviation (RSD) of parallel samples under variant LIBS parameters were analyzed (Fig 3). The large SBR represents good signal quality, and the small RSD represents good signal stability. When the laser energy was increased from 50 mJ to 90 mJ, the SBR of Te lines firstly increased, then decreased. When the laser energy was set to 70 mJ, most of Te lines obtained largest SBR and smallest RSD (Fig 3a). At lower laser energy, the plasmas might not be effectively generated, leading to weaker and more unstable signals. Conversely, at high laser energy, excessive laser energy might lead to overly aggressive plasma excitation, and the plasmas could become overly hot and dense, leading to rapid expansion and potential signal distortion, reducing signal intensity and introducing instability. At a moderate laser energy (70 mJ), the plasma temperature and electron density are optimized, allowing for the generation of sufficiently intense and stable signals. When the delay time was set to 2.0 μs, most of the Te lines obtained the largest SBR and relatively small RSD (Fig 3b). A short delay time captured the signal during the initial stage of plasma formation, leading to excessively strong and unstable signals with high background noise. While a long delay time allowed the plasmas to cool and recombine, resulting in the significant decrease of the signal intensity and stability. A properly optimized delay time (2.0 μs) ensured that the plasma had reached a more

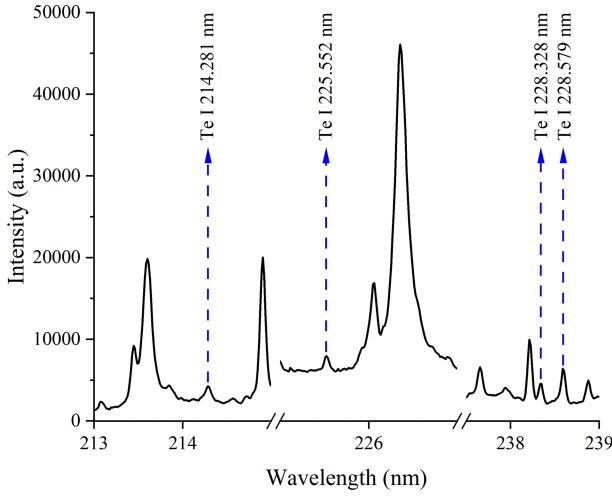

**Fig 2**. The Te lines for LIBS parameter optimization.

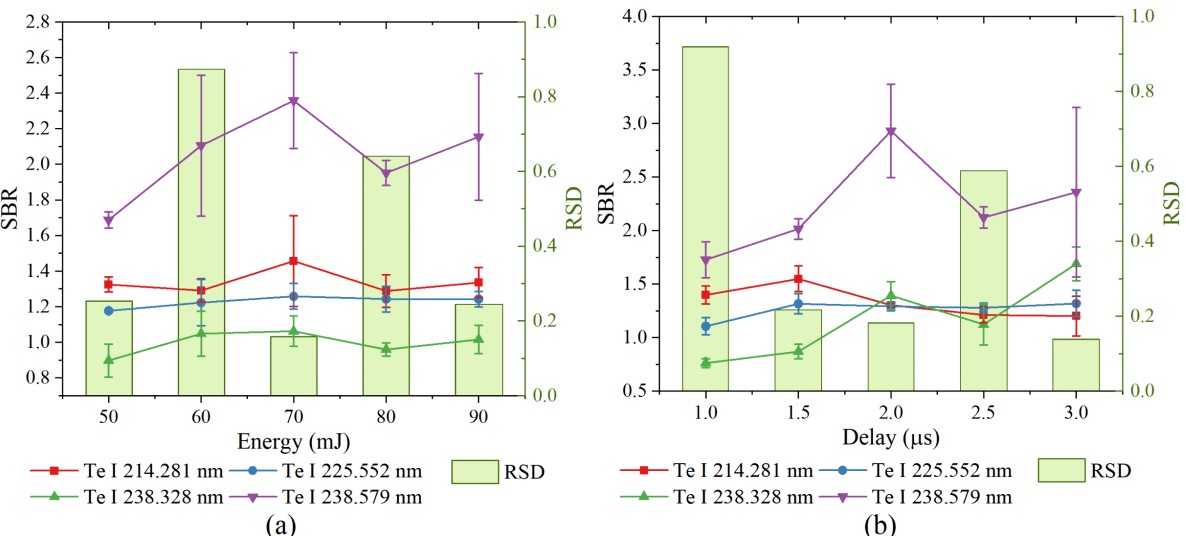

**Fig 3**. The SBR and RSD of Te lines under (a) laser pulse energy ranging from 50 to 90 mJ, and (b) delay time ranging from 1.0 to 3.0 μs.

stable phase where the signal intensity is sufficient and the background noise is minimal, allowing for high-quality and reliable signal collection. Therefore, the laser pulse energy of 70 mJ and delay time of 2.0 μs were selected as optimal LIBS parameters in the following experiments to ensure repeatability.

## Sample spectra and PCA results

The LIBS spectra of three kinds of samples were acquired and the mean spectra of them acquired by monochromator and echelle grating spectrometer were shown in Fig 4. Several spectral peaks at same wavelengths could be observed in LIBS spectra of three kinds of samples, as Cu is the major element of three kinds of materials. While the spectral lines corresponding to Te, Zn and other impurity elements (such as Fe) were of different characteristics in different kinds of samples. PCA was conducted based on the LIBS spectra of two spectroscopies, and the result was shown in Fig 5. It should be noted that the spectrum of one red copper pin was identified as an outlier and subsequently excluded from the analysis. In the PCA scatter plots based on LIBS spectra acquired by monochromator and echelle grating spectrometer, the sample points were separated into four distinct clusters, represented for tellurium copper, brass, gold-plated red copper, and gold-plated brass, respectively. It should be noticed that the brass and gold-plated brass sample were obviously separated into different clusters, and in Fig 5b, the cluster of gold-plated red copper and gold-plated brass were quite close. Though the first two times of LIBS signal for gold-plated brass pins and gold-plated red copper pins were discarded and only the last three times of the LIBS signal at each ablated location were accumulated, the gold-plated brass and brass samples were separated, and two kinds of gold-plated samples exhibited higher similarity than other two kinds of samples. As for the spectra acquired by monochromator, PC1 explained 95.98% variables, and most of the variables of high weights related to Cu lines. The difference in Cu content in variant materials lead to such discrete result. The distance between the laser focus and sample surface could cause signal intensity difference, which might lead to the separating cluster of brass and gold-plated brass. As for the spectra acquired by echelle grating spectrometer, the spectral lines related to gold (Au I 267.595 nm and Au I 242.795 nm) had larger weights in PC2, which might lead to the higher similarity of gold-plated brass and gold-plated red copper cluster on PC2 than that of the gold-plated brass and brass sample cluster (Fig 5b). The PCA results indicated that the gold plating layer could influence the LIBS spectra of plug pins, and simple PCA could not handle the classification of these materials.

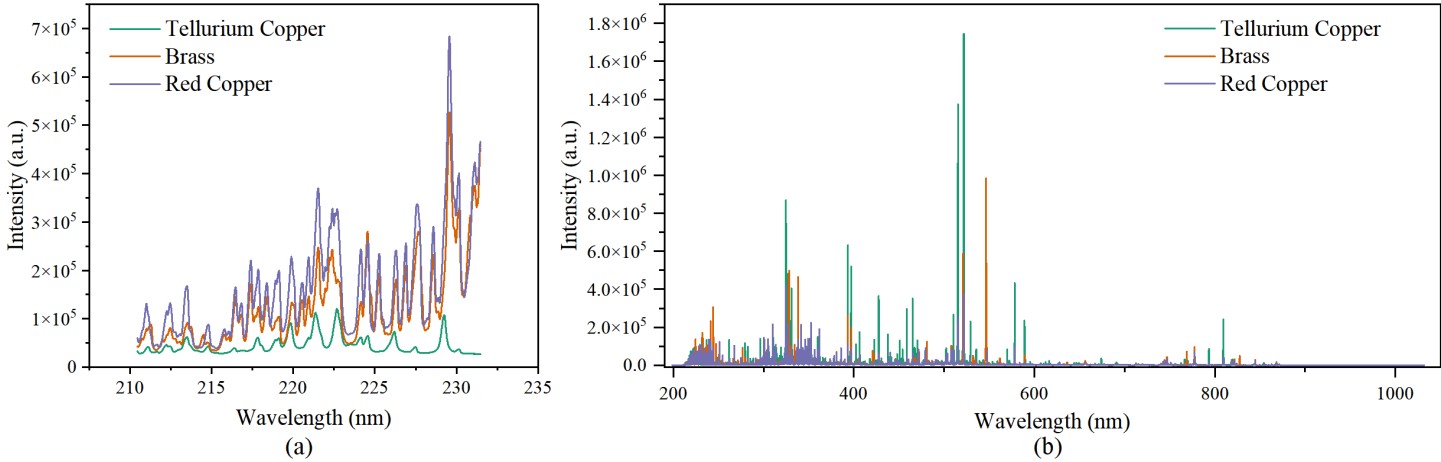

**Fig 4**. The mean LIBS spectra of Tellurium Copper, Brass and Red Copper acquired by (a) monochromator, and (b) echelle grating spectrometer.

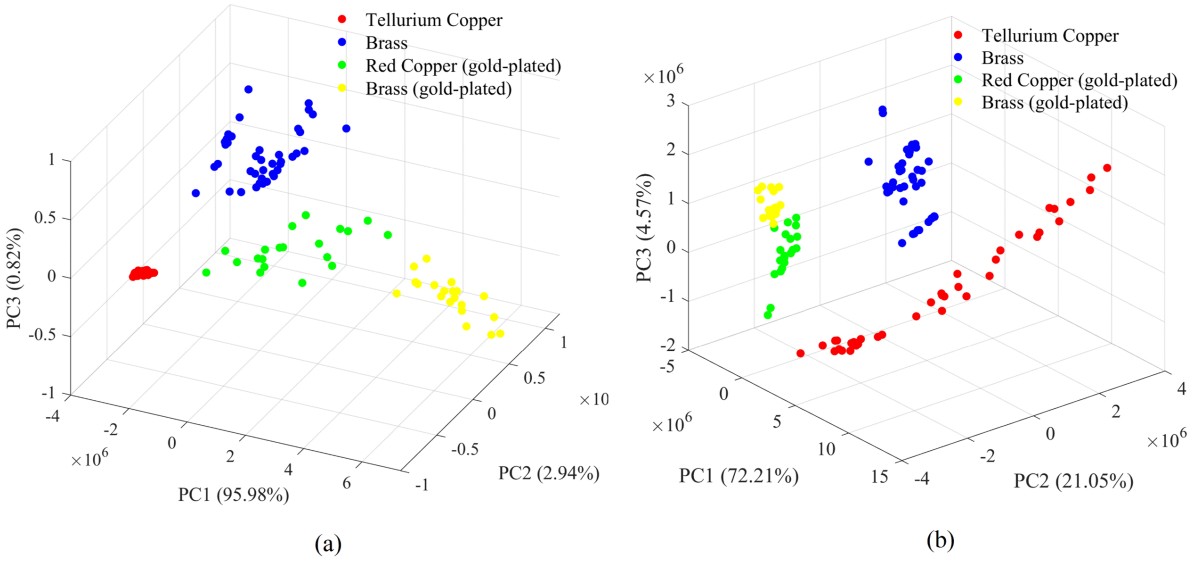

**Fig 5**. The PCA scatter plots of LIBS spectra acquired by (a) monochromator, and (b) echelle grating spectrometer.

## Discrimination models based on feature bands

To achieve convenient measurement without sample pretreatment such as removing the gold plating layer, supervised classification models were then built to improve the discrimination accuracy. To ensure the simplicity and stability of the models, the feature bands were extracted by knowledge-driven method and data-driven methods, and models were built based on different feature band datasets.

**Features extracted by knowledge-driven method.** As the Te and Zn content in three kinds of materials were obviously different (Table 1), the spectral lines related to Te and Zn had potential in discrimination of them. The Te and Zn lines listed in Table 2 were extracted as feature variables, and machine learning models, including KNN, RF and CNN models, were built based on them. The results were listed in Table 3. For the LIBS spectra acquired by echelle grating

**Table 3.** The results of machine learning models based on extracted features for discrimination of three kinds of materials.

| Spectrometer | Method[1] | Number[2] | Model | Accuracy | |
|---|---|---|---|---|---|
| | | | | **Training set** | **Testing set** |
| Echelle grating spectrometer | Te & Zn lines | 10 | KNN | 100.00% | 94.29% |
| | | | CNN | 94.05% | 85.71% |
| | | | RF | 100.00% | 94.29% |
| | SPA | 29 | KNN | 100.00% | 94.29% |
| | | | CNN | 100.00% | 94.29% |
| | | | RF | 100.00% | 97.14% |
| | CARS | 46 | KNN | 100.00% | 94.29% |
| | | | CNN | 100.00% | 94.29% |
| | | | RF | 100.00% | 91.43% |
| | Te & Zn lines + SPA | 39 | KNN | 100.00% | 97.14% |
| | | | CNN | 100.00% | 97.14% |
| | | | RF | 100.00% | 97.14% |
| | Te & Zn lines + CARS | 56 | KNN | 100.00% | 94.29% |
| | | | CNN | 100.00% | 100.00% |
| | | | RF | 100.00% | 91.43% |
| Monochromator | Te & Zn lines | 8 | KNN | 98.81% | 100.00% |
| | | | CNN | 98.81% | 100.00% |
| | | | RF | 100.00% | 100.00% |
| | SPA | 14 | KNN | 98.81% | 100.00% |
| | | | CNN | 100.00% | 97.14% |
| | | | RF | 100.00% | 100.00% |
| | CARS | 20 | KNN | 96.43% | 100.00% |
| | | | CNN | 98.81% | 100.00% |
| | | | RF | 100.00% | 100.00% |

[1]Method is the feature extraction method.
[2]Number is the number of the extracted features.

spectrometer, 6 Te lines and 4 Zn lines were extracted as feature bands, and the KNN and CNN models based on such features performed better with accuracy of 100% and 94.29% for training set and testing set, respectively. As for the LIBS spectra acquired by monochromator, 7 Te lines and 1 Zn line were extracted as feature bands, and the models based on such 8 variables performed well with accuracy of 100% for testing set. Especially the RF model based on Te & Zn lines obtained accuracy of 100% for both training set and testing set. The results indicated that the model based on Te & Zn lines extracted from spectra collected by monochromator performed better than those by echelle grating spectrometer. The sensitivity of the echelle grating spectrometer at the bands ranging from 200 nm to 240 nm was worse than the monochromator. The Te and Zn lines with relatively higher intensity were around 210 - 220 nm, but these spectral lines were not observed in the spectra collected by the echelle grating spectrometer (Table 2). Therefore, when using echelle grating spectrometer for spectra acquisition, three kind of materials could not be fully discriminated only based on these Te and Zn lines.

**Features extracted by data-driven methods.** Two data-driven feature extraction methods, SPA and CARS were utilized to selected feature bands, and KNN, RF and CNN models were built based on such feature variables. The results were listed in Table 3. The models based on features extracted from LIBS spectra acquired by echelle grating spectrometer performed closely to those based on Te & Zn lines, and the optimal model among them was the RF model based on 29 features extracted by SPA, of which the accuracy for training set and testing set were 100% and 97.14%, respectively. The models based on features extracted from monochromator also performed well, and most of them obtained accuracy

of 100% for the testing set, particularly the RF models based on features extracted by SPA or CARS, achieving accuracy of 100% for both training set and testing set. By feature selection, the variables decreased from 1024 to 14 and 20, respectively.

**Feature fusion of data-driven and knowledge-driven methods.** To improve the classification accuracy based on echelle grating spectrometer, the feature fusion of data-driven and knowledge-driven methods was conducted, and the results of models based on fusion features were listed in Table 3. The models based on fusion features of SPA and Te & Zn lines performed a little better than those only using SPA feature or Te & Zn lines, with accuracy of 100% for training set and 97.14% for testing set, respectively. The CNN model based on fusion features of CARS and Te & Zn lines performed best among models using echelle grating spectrometer features with accuracy of 100% for both training set and testing set.

The feature bands of spectra collected by echelle grating spectrometer and monochromator by knowledge driven method and data-driven methods were shown in Fig 6. As shown in Fig 6a, for the spectra collected by monochromator, the SPA features and Te & Zn lines exhibited a high degree of similarity, while the CARS features showed significant differences from the other two methods. The SPA algorithm tends to selected variables that maximize the interclass differences, and as Te and Zn were elements different in three kinds of materials, Te lines and Zn lines were selected by SPA. While CARS selected features based on their importance in PLS regression model, and Cu was the major element with different content in three materials, Cu lines were mainly selected as CARS feature. The RF models based on three feature datasets all performed well.

As for the spectra collected by echelle grating spectrometer (Fig 6b), the SPA features still included most of the Te lines and Zn lines, and included some Cu lines as well. The CARS algorithm also focused more on feature bands related to Cu, including Cu I 324.7537 nm, Cu I 510.5537 nm, Cu I 521.8197 nm and Cu I 515.323 nm. Some bands related to impurity elements, such as nickel (Ni I 352.454 nm) and iron (Fe I 547.6564 nm), were selected as features as well. The models based on three feature datasets did not performed well as those based on spectra acquired by monochromator. By fusing the features extracted by data-driven and knowledge-driven methods, the accuracy of most models improved, indicating that spectral lines related to Cu, Te, Zn and some impurity elements played a key role in charging plug material discrimination.

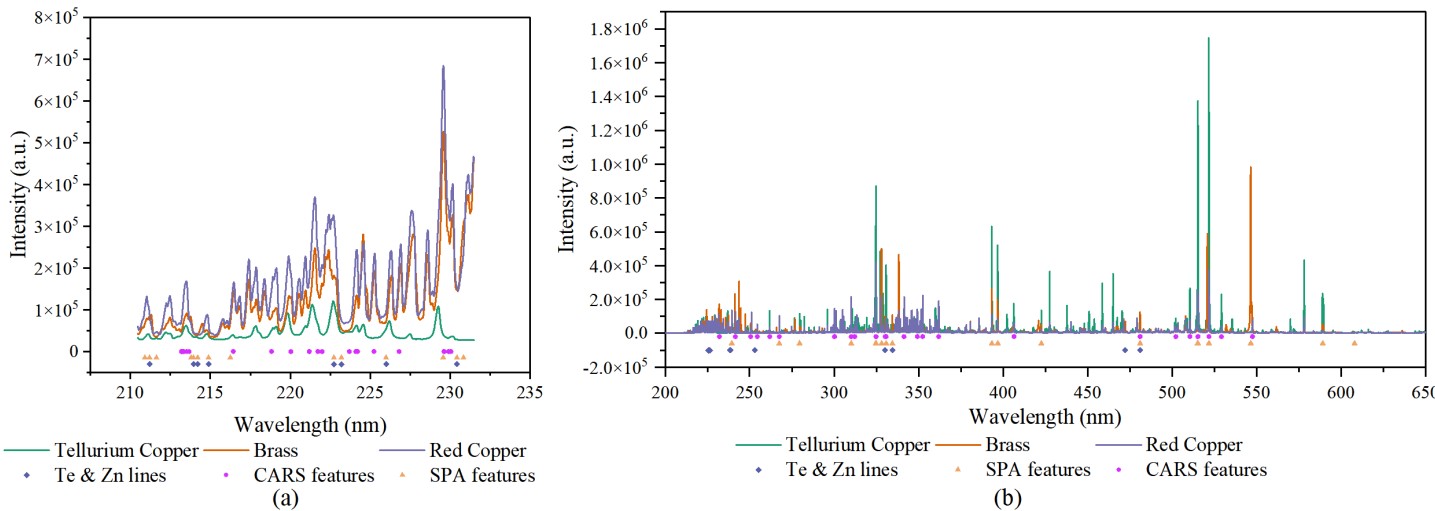

**Fig 6**. **The feature bands extracted by CARS, SPA algorithm and Te & Zn lines extracted based on NIST database from spectra acquired by (a) monochromator, and (b) echelle grating spectrometer.**

It should be noted that the self-absorption occurs when emitted light is absorbed by surrounding atoms or ions, leading to a reduction in the intensity of spectral lines. This effect is particularly pronounced for high-concentration elements or strong emission lines. Since Cu was the main element in three kinds of materials (Table 1), Cu I 324.7537 nm and Cu I 327.3954 nm lines in all samples were affected by self-absorption, with their intensities being significantly lower than the theoretical values. And Zn, as the second most abundant element in brass, (Table 1), caused a mild self-absorption effect on Zn I 213.8573 nm lines in brass samples. While other weaker Cu and Zn lines, as well as the spectral lines associated with elements of low concentration (e.g., Te, Fe, Ni), were not notably impacted by self-absorption under the optimized experimental conditions. Despite this, the Cu I 324.7537 nm and Zn I 213.8573 nm lines were still selected as features by both the CARS and SPA methods (Fig 6), and the models built using these features performed well (Table 3). Given that the self-absorption effect on strong Cu lines affected all three kinds of samples, and the impact on strong Zn lines primarily affected brass samples, the overall influence of self-absorption on materials discrimination can be considered negligible.

### Discrimination models based on full-spectra

The KNN, RF and CNN models were also built for discrimination of three kinds of materials based on the full-spectra acquired by echelle grating spectrometer and monochromator, respectively, and the results were listed in Table 4. As the full spectra contained the information of most elements, all models performed well with the accuracy of testing set higher than 94%, particularly the CNN model based on full monochromator spectra and KNN model based on echelle grating spectrometer spectra, which obtained accuracy of 100% for both training set and testing set. The results based on full-spectra indicated that LIBS could correctly discriminate tellurium copper, brass and red copper based on the metal-related spectral lines, and the spectral lines related to gold-plating layer and other irrelevant elements would not interfere the classification.

The results of the models based on feature bands and full spectra indicated that both monochromator and echelle grating spectrometer could realize the precise and fast identification of the three kinds of charging pile plug materials, providing rich spectrometer options for the application in industry. And the gold plating layer on the surface of them will not interfere with the classification, ensuring the convenience of the detection.

### Conclusion

This study introduced a rapid charging pile plug material identification method based on LIBS to discriminate the tellurium copper from two commonly used plug materials. Two kinds of charging plugs, gold-plated red copper and gold-plated brass, and tellurium copper pellet were detected by LIBS system, and two spectrometers, an echelle grating spectrometer and a monochromator, were utilized to collected spectral signal. The Te signals were identified and LIBS parameters for tellurium copper detection was optimized. The optimal laser energy and delay time were 70 mJ and 2.0 µs, respectively. The KNN, RF, and CNN models were built based on feature bands and full spectra for classification of three kinds of materials. Knowledge-driven and data-driven methods were used for feature bands extraction, and classification models

**Table 4**. **The results of machine learning models based on full-spectra for discrimination of three kinds of materials.**

| Model | Accuracy | | | |
|---|---|---|---|---|
| | Monochromator | | Echelle grating spectrometer | |
| | Training set | Testing set | Training set | Testing set |
| KNN | 97.62% | 94.29% | 100.00% | 100.00% |
| RF | 100.00% | 94.29% | 100.00% | 97.14% |
| CNN | 100.00% | 100.00% | 100.00% | 97.14% |

based on them were built. The RF models based on features extracted by SPA, CARS or Te & Zn lines from monochromator spectra achieved accuracy of 100% for training set and testing set. The CNN model based on the fusion feature extracted by CARS and Te & Zn lines from echelle grating spectrometer also achieved accuracy of 100% for training set and testing set. The spectral lines related to Cu, Te and Zn played key roles in the discrimination. The CNN model based on full monochromator spectra and the KNN model based on full echelle grating spectrometer spectra both performed well with accuracy of 100% for training set and testing set. Both two spectrometers could realize the precise and fast identification of the three kinds of plug materials without pretreatments. The proposed LIBS-based rapid charging pile plug material discrimination method could provide references for improving the quality of electric vehicle charging infrastructure, improving the efficiency, safety, and sustainability of electric vehicle charging plug production.

## Acknowledgments

This study was supported by Zhejiang Key Laboratory of Agricultural Remote Sensing and Information Technology.

## Author contributions

**Conceptualization:** Lidan Chen, Fei Liu.

**Data curation:** Tiantian Pan.

**Formal analysis:** Lidan Chen.

**Funding acquisition:** Fei Liu.

**Investigation:** Lidan Chen, Tiantian Pan, Liuye Cao.

**Methodology:** Lidan Chen, Fei Liu.

**Project administration:** Lidan Chen.

**Resources:** Lidan Chen.

**Software:** Tiantian Pan, Liuye Cao.

**Supervision:** Fei Liu.

**Validation:** Liuye Cao.

**Visualization:** Tiantian Pan.

**Writing – original draft:** Lidan Chen, Tiantian Pan.

**Writing – review & editing:** Fei Liu.

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
