## [Decision Letter · Decision Letter 0]

29 Dec 2025

PONE-D-25-49098Fast identification of the charging pile plug materials using laser-induced breakdown spectroscopyPLOS One

Dear Dr. Pan,

Thank you for submitting your manuscript to PLOS ONE. After careful consideration, we feel that it has merit but does not fully meet PLOS ONE’s publication criteria as it currently stands. Therefore, we invite you to submit a revised version of the manuscript that addresses the points raised during the review process.

We look forward to receiving your revised manuscript.

Kind regards,

Talaat Abdel Hamid, Ph.D

Academic Editor

PLOS One

“This study was supported by the Research Funding of Zhejiang Key Laboratory of Agricultural Remote Sensing and Information Technology”

6. Please amend either the abstract on the online submission form (via Edit Submission) or the abstract in the manuscript so that they are identical.

Reviewers' comments:

Reviewer's Responses to Questions

**Comments to the Author**

1. Is the manuscript technically sound, and do the data support the conclusions?

Reviewer #1: Yes

2. Has the statistical analysis been performed appropriately and rigorously?

Reviewer #1: Yes

3. Have the authors made all data underlying the findings in their manuscript fully available?

Reviewer #1: No

4. Is the manuscript presented in an intelligible fashion and written in standard English?

Reviewer #1: Yes

5. Review Comments to the Author

Reviewer #1: Plasma condition clarification

Although this study focuses on classification rather than quantitative analysis, a brief discussion of plasma stability or repeatability (e.g., justification of optimized laser energy and delay time in relation to plasma behaviour) would improve the physical completeness of the work.

Self-absorption consideration

Several strong Cu, Zn, and Te emission lines are used for feature extraction. A short statement discussing whether self-absorption effects are expected to be negligible under the selected experimental conditions would enhance the robustness of the interpretation.

Model validation clarity

The reported classification accuracies are very high, in several cases reaching 100%. Clarifying whether additional validation (e.g., repeated random splits or cross-validation) was tested, or adding a brief discussion on model generalizability, would strengthen the statistical interpretation without requiring major reanalysis.

Data availability

The Data Availability Statement currently indicates that the data are not available. To comply with PLOS ONE data policy, the authors should provide representative raw spectra or feature datasets as Supporting Information or through a public repository.

Presentation improvements

Minor improvements are recommended for figure clarity (axis labels, units) and correction of small grammatical or typographical issues throughout the manuscript.

Overall Recommendation

The manuscript is technically sound and addresses a relevant application of LIBS. The suggested revisions are minor and mainly intended to improve transparency, clarity, and compliance with journal policies. I therefore recommend minor revision.

6. PLOS authors have the option to publish the peer review history of their article (what does this mean?). If published, this will include your full peer review and any attached files.

Reviewer #1: **Yes:** Ambreen Aslam

---

## [Author Response · Author response to Decision Letter 1]

5 Jan 2026

Response to Reviews

We thank reviewer for thoughtful and constructive comments on our manuscript. We greatly appreciate the time and effort you have dedicated to reviewing our work. Your detailed feedback has been incredibly helpful in improving the clarity and quality of our paper. We have carefully revised the manuscript, and we hope that the changes made address your concerns. Below, we provide a point-by-point response to your suggestions and questions.

Reviewer #1:

1. Plasma condition clarification

Although this study focuses on classification rather than quantitative analysis, a brief discussion of plasma stability or repeatability (e.g., justification of optimized laser energy and delay time in relation to plasma behaviour) would improve the physical completeness of the work.

Response: Thanks for your insightful feedback. We have added the discussion of the effect of laser energy and delay time on plasma stability in Lines 202-218. At a moderate laser energy, the plasma temperature and electron density are optimized, allowing for the generation of sufficiently intense and stable signals. And a properly optimized delay time ensured that the plasma had reached a more stable phase where the signal intensity is sufficient, and the background noise is minimal, allowing for high-quality, reliable and stable signal collection.

2. Self-absorption consideration

Several strong Cu, Zn, and Te emission lines are used for feature extraction. A short statement discussing whether self-absorption effects are expected to be negligible under the selected experimental conditions would enhance the robustness of the interpretation.

Response: Thanks for your valuable suggestion. We have added the discussion on self-absorption effects in Lines 317-332. We have found that the Cu I 324.7537 nm and Cu I 327.3954 nm lines in all samples, and the Zn I 213.8573 line in brass samples were impacted by self-absorption due to their high concentration. While other weaker Cu and Zn lines, as well as the spectral lines associated with elements of low concentration (e.g., Te, Fe, Ni), were not notably impacted under the optimized experimental conditions. Despite this, the strong Cu I 324.7537 nm and Zn I 213.8573 nm lines were selected as features by both the CARS and SPA methods, and the models built using these features performed well. Given that the self-absorption effect on strong Cu lines affected all three kinds of samples, and the impact on strong Zn lines primarily affected brass samples, the overall influence of self-absorption on materials discrimination can be considered negligible.

3. Model validation clarity

The reported classification accuracies are very high, in several cases reaching 100%. Clarifying whether additional validation (e.g., repeated random splits or cross-validation) was tested, or adding a brief discussion on model generalizability, would strengthen the statistical interpretation without requiring major reanalysis.

Response: Thanks for your thoughtful feedback. The reported classification accuracies in the study are based on a single split of the data into training and test sets, without the use of additional validation techniques such as repeated random splits or cross-validation. We have also added the detail in Lines 152-153 and Lines 161-162.

4. Data availability

The Data Availability Statement currently indicates that the data are not available. To comply with PLOS ONE data policy, the authors should provide representative raw spectra or feature datasets as Supporting Information or through a public repository.

Response: Thanks for your valuable feedback. We have made the full spectra and feature datasets available through a public GitHub repository. The dataset and code can be accessed at the following link: https://github.com/ptthoshi-web/LIBS-Charging-pile-plug. We have updated Data Availability statement (Lines 380-381) as well.

5. Presentation improvements

Minor improvements are recommended for figure clarity (axis labels, units) and correction of small grammatical or typographical issues throughout the manuscript.

Response: Thanks for your kind suggestion. We have made the recommended minor improvements to the Fig. 2, 3, 4, and 6 for clarity. Additionally, we have carefully reviewed the manuscript and corrected the grammatical and typographical issues throughout (marked in blue).

---

## [Editor Report · Decision Letter 1]

19 Jan 2026

Fast identification of the charging pile plug materials using laser-induced breakdown spectroscopy

PONE-D-25-49098R1

Dear Dr. Pan,

We’re pleased to inform you that your manuscript has been judged scientifically suitable for publication and will be formally accepted for publication once it meets all outstanding technical requirements.

Kind regards,

Talaat Abdel Hamid, Ph.D

Academic Editor

PLOS One
---

## [Editor Report · Acceptance letter]

PONE-D-25-49098R1

PLOS One

Dear Dr. Pan,

I'm pleased to inform you that your manuscript has been deemed suitable for publication in PLOS One. Congratulations! Your manuscript is now being handed over to our production team.

Kind regards,

on behalf of

Dr. Talaat Abdel Hamid

Academic Editor

PLOS One